Accessible interactive learning of mathematical expressions for school students with visual disabilities

http://orcid.org/0000-0002-7734-7243 Ali Amjad 1 amjad.cs@uop.edu.pk
Khusro Shah 1
http://orcid.org/0000-0002-0067-692X Alahmadi Tahani Jaser 2
1 Department of Computer Science, University of Peshawar , Peshawar , Pakistan
2 Department of Information Systems, College of Computer and Information Sciences, Princess Nourah bint Abdulrahman University , Riyadh , Saudi Arabia
Alatas Bilal
Electronic publication date: 2024 Dec 23
Publication date: 2024
Volume: 10
Electronic Location ID: e2599
Received 2024 Jul 4; Accepted 2024 Nov 20
Copyright: © 2024 Ali et al.
Copyright year: 2024
Copyright holder: Ali et al.
License: This is an open access article distributed under the terms of the Creative Commons Attribution License, which permits unrestricted use, distribution, reproduction and adaptation in any medium and for any purpose provided that it is properly attributed. For attribution, the original author(s), title, publication source (PeerJ Computer Science) and either DOI or URL of the article must be cited.
License URL: https://creativecommons.org/licenses/by/4.0/

Keywords: Inclusive learning, Math accessibility, Math problem solving, Navigation, Visual disability

Funding: University of Peshawar supported this research and King Salman Center for Disability Research funded this through Research Group KSRG-2023-021 University of Peshawar supported this research and King Salman Center for Disability Research funded this through Research Group No. KSRG-2023-021. There was no additional external funding received for this study. The funders had no role in study design, data collection and analysis, decision to publish, or preparation of the manuscript.

==============================
Globally, students with visual disabilities face significant challenges in accessing and learning mathematics, particularly when solving mathematical equations and expressions. These challenges result from the inherent complexity and abstract nature of mathematical content. Additionally, braille codes are inconsistent across regions, collaborative math platforms are unavailable, and accessible mathematics literature is scarce. Assistive technologies, artificial intelligence, and educational resources have improved accessibility for students with visual disabilities. However, these students still face significant challenges when navigating, exploring, and solving mathematical equations and expressions. These challenges contribute underrepresentation of these students in the science, technology, engineering, and mathematics disciplines. To address these limitations, this study proposes a novel solution to assist students with visual disabilities in learning mathematical expressions interactively with flexible navigation. This study proposes an algorithmic approach for converting input mathematical expressions into content MathML expressions, parsing those expressions into semantic elements, and then providing a structural overview of these expressions. Moreover, interactive keyboard keys were designed to provide flexible navigation through speech feedback, so that users can interact more effectively with expressions. Python libraries were utilized to implement the proposed solution. An empirical evaluation was conducted by 15 instructors and 94 students with visual disabilities and validated by Cronbach’s alpha. Results indicate that the proposed solution improved mathematics accessibility and learning. This study lays a foundation for future research on the integration of advanced technologies in special education.

Introduction

Globally, 2.2 billion people suffer from near or distance vision impairment, including 188.5 million who have mild sight impairment, 217 million who have moderate to severe sight impairment, 36 million who are blind, and 826 million with near vision impairment (WHO, 2024). In the years ahead, it is predicted that this figure will increase (Ackland, Resnikoff & Bourne, 2017; Chen et al., 2024). To provide inclusive learning opportunities to students with visual disabilities, a worldwide educational institute was established. Despite the challenges inherent in the learning process, a substantial dropout rate prevails among this demographic (Mejía et al., 2021b).

Across all disciplines, mathematics is a fundamental building block for academic success. In high school, mathematics is a baseline for social and professional success (Jariwala, 2022). In the realm of accessibility, MathML and LaTeX stood out as pivotal tools, integral to rendering mathematical content more accessible, particularly for users reliant on screen readers (Ahmetovic et al., 2021; Kruger, De Wet & Niesler, 2023). Although these languages enable precise interpretation and vocalization of math content, they have inherent limitations that prevent students with visual disabilities from independently learning, problem-solving, teaching, and navigating mathematics (Maćkowski et al., 2018). Mathematical expressions and equations are difficult for the students to comprehend because they lack an equivalent to the graphical representation that aids sighted individuals in immediate comprehension, while visual representation is immediate, audio and braille require repetitive repetition and lack immediacy (Jitngernmadan et al., 2017).

These challenges have been heightened through digital learning and inclusive education. Students with visual disabilities have difficulty comprehending mathematical expressions (mathematical expressions) and need additional support to grasp the intricacies of mathematical concepts (Ketema Dabi & Negassa Golga, 2023). Various tools and methods allow for different levels of accessibility to mathematical content (Al-Salman, 2008; Fajardo-Flores & Archambault, 2012). However, these tools and methods do not sufficiently represent mathematical expressions and their underlying structures to facilitate manipulation. As a consequence, conventional formats such as bitmap images and PDF documents perpetuate inaccessibility (Fayyaz, Khusro & Imranuddin, 2023; Tarida, 2022; ZafarIqbal et al., 2021).

A growing number of researchers are finding ways to develop software that facilitates interaction with mathematical expressions (mathematical expressions). Despite significant progress in promoting inclusivity in mathematics education, it remains difficult to access, navigate, and comprehend Mathematical expressions (Maćkowski et al., 2023a; Shoaib et al., 2023). Digital talking books (DTB) and digital accessible information systems (DAISY) are both based on MathML to represent mathematical expressions but lack comprehensive navigation and audio descriptions (Kerscher, 2001; Leas et al., 2008). MathPlayer offers MathML support, but there are compatibility issues and semantic navigation limitations (Raman & Gries, 1995; Soiffer, 2005; Stevens & Edwards, 1994). Even though screen readers like NVDA (Bigham, Prince & Ladner, 2008) and JAWS (Sandhya & Devi, 2011) provide navigation modes, they fail to provide semantic structure comprehension form. Despite the abstract hierarchical nature of mathematical expressions, existing tools often fail to provide an adequate elucidation of their structural framework, which includes terms, factors, constants, and variables, that are essential for equation resolution (da Paixão Silva et al., 2018; Miesenberger et al., 2023).

This study presents a novel solution tailored to address accessibility, navigation, and comprehension challenges inherent in mathematical expressions for students with visual disabilities. By examining mathematical expressions from different perspectives, we untangle their hierarchical structure, revealing sub-elements, terms, factors, constants, and variables. Using the proposed interactive key, students can investigate the expressions, to grasp more detailed information about their meaning and structure.

This research proposes a novel algorithmic solution that can help the learning of mathematics curriculum for students with visual disabilities in the 6th to 10th grades. It is focused on enhancing the concept of teaching and learning troubles of algebraic linear and quadratic mathematical expression with the innovation of teaching approaches. The objective is to develop teaching and learning processes, adopting an intuitive understanding of these essential mathematical concepts. This study focused on the math curriculum in the Oxford and Khyber Pakhtunkhwa textbooks, maintaining their inherent approach to mathematical problem-solving. In contrast to traditional formats such as Latex and MathML, our algorithmic framework facilitates problem-solving approaches, rather than serving as representations of mathematical expressions.

An empirical evaluation based on qualitative and quantitative analysis of the proposed solution was carried out with 15 instructors and 94 students with visual disabilities and has demonstrated that it significantly improves accessibility and learning of mathematical equations and expressions. The discussion addresses issues related to structural information, semantic descriptions, and navigation in assistive technologies and eLearning tools. This paper contributes to improving the accessibility of education by proposing a new approach to help students with visual disabilities better understand mathematical content. The paper is organized as follows: ‘Related Work’ provides a literature review, ‘Proposed Solution’ presents the proposed solution, ‘Materials and Methods’ describes the materials and methods, ‘Results and Discussion’ covers the results and a discussion, and the ‘Conclusion’ concludes this article.

Related work

Mathematical expressions and equations are difficult to understand and manipulate for students with visual disabilities (Ali & Khusro, 2024). While several assistive technologies have been developed to address these challenges, each has its limitations, particularly in terms of navigation, manipulation, and semantics (Al Shehri et al., 2022). This section reviews relevant work in six key areas critical to addressing these challenges: screen reader technologies, braille and other electronic formats, computer algebra systems, artificial intelligence tools, the applicability of the LLMs for mathematical education, a comparison of the proposed tool with the closest counterparts in respect to navigation and structure. Each proposed subsection contains a more detailed analysis of the approaches and tools discussed and emphasizes the existing gaps in the tools currently available today as well as the comparison with the proposed approach.

Screen readers technology

Screen reader technology converts the on-screen math text into synthesized speech. Despite its importance, screen reader technology serves limited purposes when it comes to interpreting and presenting mathematical expression structure (da Paixão Silva et al., 2017). There are many mathematical symbols, such as superscripts, subscripts, and specialized characters, that pose a major challenge to screen readers because they may have difficulty conveying these elements accurately, resulting in incomplete or inaccurate auditory presentations (Maćkowski et al., 2018). As a result, screen reader output does not capture the hierarchical and spatial relationships inherent in mathematical expressions, such as fractions or matrices, which depend on the spatial layout to be understood (Miesenberger et al., 2022). There are challenges associated with tools that convert MathML and LaTeX into accessible formats, such as formatting inconsistencies and incomplete support for advanced mathematical expressions (Greiner-Petter, 2023; Klingenberg, Holkesvik & Augestad, 2020). These tools are Mathjax, NVDA With Math player, JAWS, Voice over, and Speech Rule Engine (SRE) (Mejía et al., 2021a). However, those tools offer the straightforward generation of Math-ML representations of the mathematical formulas in the formats of MathML, but they have some disadvantages, for instance, showing the structure of the document and having changeable navigation (Maćkowski et al., 2023b). As a consequence, this study proposed a solution to address these challenges to making mathematical expression more accessible, present the structure overview of math expressions and provide flexible navigation to understand mathematical expressions.

MathML and LaTeX-based tools

MathML is one of the W3C recommendations that offer a clear representation of a mathematical notation, as well as making the notations machine-readable, so a screen reader can translate it into speech or braille (Manzoor et al., 2019). The conversion of MathML and LaTeX-based content into accessible formats requires intricate formatting steps. According to Raman & Gries (1995), Soiffer (2005), Stevens & Edwards (1994), Math Player converts MathML content into a voice for the browser. Although Math Player is useful, it has limitations, such as limited navigation and manipulation features, mobile support, and platform support (Al-Razgan et al., 2021). Wang et al. (2021) also describe converting LaTeX and TeX code to MathML. However, MathML and LaTeX-based tools can present accessibility challenges due to formatting requirements (Schmitt-Koopmann et al., 2024).

Using MathML support in screen readers, students with visual disabilities can perceive mathematical concepts in a machine-readable format by converting MathML code into speech or braille (Miesenberger et al., 2022). In addition to syntax highlighting and audio feedback, students can use a MathML editor with accessibility features to create and modify MathML code (Frankel, Brownstein & Soiffer, 2017). Additionally, LaTeX-to-MathML conversion tools address the challenge of existing LaTeX content by converting it into accessible MathML format (Stamerjohanns et al., 2009). Interactive platforms that support MathML provide these students with enhanced learning opportunities, increased independence in math education, and better access to mathematical content (Brzostek-Pawłowska, Rubin & Salamończyk, 2019). MathML and LaTeX are used to create these tools (Greiner-Petter, 2023). However, MathML still faces challenges such as limited adoption by educators and publishers, possible complexity of MathML code, and the need for ongoing development to keep pace with technological advancements and evolving pedagogy (National Research Council, Division on Engineering and Physical Sciences, Board on Mathematical Sciences and Their Applications, Committee on Planning a Global Library of the Mathematical Sciences, 2014). There are several directions underway in the future, including standardizing and integrating MathML, developing user-friendly MathML authoring tools, and developing pedagogical approaches to maximize the learning potential of these tools (Thai, Bang & Li, 2022). Our proposed solution offers technological approaches to this limitation to assist students with visual disabilities in learning mathematical expressions.

Braille and E-book formats

Math contents can be presented in braille through refreshable braille displays which enable students with visual disabilities to have touch-friendly content (Russomanno et al., 2015). However, many of these displays can only display simple mathematical expressions and require braille proficiency, so they are not widely used (Brzostek-Pawłowska, Rubin & Salamończyk, 2019). Students with visual impairments can access literary content more easily with the DAISY format, which has structured navigation, text-to-speech, and compatibility with assistive technology. For inclusion education and equal access to literature, irrespective of print disabilities, it is essential to address challenges and spread awareness of the DAISY format (Akbar et al., 2024). The DAISY format, however, is limited in its ability to display complex mathematical notation, navigate equations seamlessly, and provide tactile representation, which requires additional assistance (Ali & Khusro, 2024). Our proposed solution is user-friendly, requires no additional training, and is cost-effective.

Computer algebra systems

Mathematical problems, especially algebraic ones, can be solved with advanced tools like computer algebra systems (CAS), but they can be difficult to access for students with visual disabilities (Mejía et al., 2021b). Complex mathematical computations are handled efficiently by popular CAS platforms like Mathematica, Maple, and Maxima (Shingareva & Lizárraga-Celaya, 2010). Further, these systems generally lack navigation capabilities, have inaccessible interfaces, have steep learning curves, and have inaccessible graphic outputs. Users with severe visual disabilities are often unable to access their outputs. Similarly, IrisMath, a web-based computer algebra system, addresses mathematical problem-solving needs but faces its own set of challenges (Zambrano et al., 2023). Input mechanisms are complex, and navigation and analysis features are limited, making it hard to use. Although Mathematica, Maple, Maxima, and IrisMath are powerful computational tools, their usability and accessibility issues make them unappealing to a wide audience (Maćkowski et al., 2023a).

AI-based tools for math accessibility

The challenge of mathematics for students with visual impairment persists especially in teaching-learning environments that are heavily invested with visualization (Lemessa, 2023). However, the landscape of these technologies changed in recent years due to new advancements in artificial intelligence (AI) and large language models (LLMs) which seek to provide new solutions that could improve accessibility (Kshetri, 2024). Computer vision technologies including optical character recognition (OCR) and speech-to-text translation can fill the existing gap left by most learning methods between conventional and innovative experience-based approaches (Sakshi & Kukreja, 2024). OCR technologies assist in digitizing texts of mathematical print so that the screen readers can easily translate them and make them more easily navigable for a student; speech recognition allows students to solve math problems verbally making the learning process more engaging (Luo et al., 2024).

Similarly, current state-of-art LLMs like OpenAI’s GPT-4, have shown the ability to both generate and comprehend mathematical content. It must be noted that some of these models can turn mathematical computations into the normal language to enhance learning for these students (Collins et al., 2024). In addition, LLMs like ChatGPT can solve math problems in several steps, come up with a second version of the explanation or explanation in their own way, and adapt the answers to a particular learner’s needs and preferences, which makes them useful tools in learning personalization contexts (Alto, 2023). But despite that, there are disadvantages of LLMs including interaction with non-text content, high wordiness, and accessibility issues for users with disabilities via screen readers and braille devices displays (Kuzdeuov et al., 2024; Sohail et al., 2023). Despite their potential, artificial intelligence-based tools still have critical issues in interaction and navigation, limiting their effectiveness (Mejía et al., 2021a). These limitations are mitigated in our proposed system as it provides non-redundant interaction and compliance with screen reader-based interfaces making for a far more efficient experience for the students with visual disabilities.

Analysis of tools and studies supporting mathematical expressions

To make mathematics accessible for students with visual disabilities, several technological tools are available, however, they differ in their features and accessibility. Our proposed solution is compared to existing tools in Table 1 below, emphasizing key features such as structural overview, elements identification, and navigation.

Table 1 Analysis of tools and studies supporting mathematical expression.

Tool/Study	Accessible equation structure overview	Equations or expression elements identification	Navigation	Screen-reader compatibility	Limitations	Empirical evaluation	
MathPlayer (Raman & Gries, 1995; Soiffer, 2005; Stevens & Edwards, 1994)	MathML base structure	Basic identification	Minimal linear reading only	Yes	But MathPlayer is a verbose app that is confined to web use only and creates illicit ‘start’ and ‘stop’ cues that make it inefficient to comprehend.	Not empirically evaluated	
REMAthEx (Gaura, 2002)	MathML base structure	Basic identification	Minimal linear reading only	Yes	REMathEx is only applicable to the basic expressions, MathML calls, basic editing/& navigation, and does not support a wider selection of braille standards, thus reduces its flexibility and able to use.	Not empirically evaluated	
Math Genie (Gillan et al., 2004)	Provides left-to-right structure overview based on sighted users’ reading patterns	Identifies elements, including parentheses and operators, allowing chunking and substitution	Enables navigation to specific points, including start and within parentheses	Yes	Lack of structured feedback or equation view	Empirically evaluated	
AudioMath (Sánchez & Flores, 2005)	Lacks a detailed structural overview of equations	Limited identification of individual elements	Basic linear navigation only	Yes	Lacks structured feedback so users cannot interpret relationships within expressions.	Not Empirically evaluated	
MatVOX (Sanmiguel & Martini, 2012)	Limited to basic and simple athematic problem	Limited identification of individual elements	No	No	Limited in dealing with complex equations	Empirically evaluated	
L-Math (Barbieri, Mosca & Sbattella, 2008)	Simple reading	No	No	No	No structured feedback, expression relationships not easily expressed, no complex equation navigation.	Empirically evaluated	
LAMBDA (Schweikhardt et al., 2006)	Provides linear, braille-compatible structure through LAMBDA code	Identifies elements like fractions and operators with structured tags	Supports precise navigation to specific tags and substructures	Yes, compatible with braille and audio synthesis for speech output	Limited to linear representations, lacks support for complex visual layouts, and requires integration with mainstream software for wider use.	Empirically evaluated	
i-Math (Wongkia, Naruedomkul & Cercone, 2012)	Provides basic audio overview of math expressions	Identifies math elements for TTS (Text-to-Speech) output	No	No	Lacks support for PDF and bitmap-based math expressions; limited interactivity for complex math content	Empirically evaluated	
Mathspeak (Sheikh, Schleppenbach & Leas, 2018)	Provides detailed verbal description of equations, emphasizing hierarchical structure	Identifies elements such as fractions, roots, integrals, subscripts, and superscripts	Allows navigation through different sections and blocks of equations	Compatible with TTS systems for LaTeX and MathML input	Verbose output may overwhelm users with complex expressions, requiring cognitive effort to follow lengthy description	Empirically evaluated	
EAR-Math (Kacorri, Riga & Kouroupetroglou, 2014)	Utilizes a detailed audio rendering of structural elements, operators, and numbers with error-based metrics	Recognizes and tracks errors in elements like structural components, arithmetic operators, and identifiers	Allows repeated attempts with incremental error correction	Compatible with MathPlayer and other TTS system	High initial error rate for complex structures, with manual alignment required for metric calculation	Empirically evaluated	
JAWS (Job Access With Speech) (Bankar & Lihitkar, 2022), NVDA-Nonvisual Desktop Access (NV Access, 2017)	NVDA provides linear audio description of equations when paired with MathPlayer or MathCAT	Identifies elements like fractions, exponents, and roots with accessible labels	Basic linear navigation, with limited exploration of math structure	Compatible with Windows and supports MathPlayer, MathCAT, and Access8Math	Limited support for complex math layout; navigation can be challenging for lengthy expressions; lacks native braille support for advanced math notations	Not empirically evaluated	
MathType (Hindin, 2010)	Provides a visual editor for structured equations, displaying components like fractions and exponents	Identifies elements through WYSIWYG interface, supporting LaTeX and MathML	Limited internal navigation within expressions	Compatible with NVDA for basic reading, requires MathPlayer for audio support	Subscription required; limited screen reader compatibility without MathPlayer; files can be large and lack portability with some assistive tools	Not empirically evaluated	
MathMelodies (Cantù et al., 2018)	Uses accessible audio descriptions and interactive exercises in a structured storyline format	Identifies interactive elements (e.g., counting icons) with audio feedback	Provides gesture-based navigation compatible with VoiceOver and TalkBack	Compatible with iOS and Android screen readers; limited advanced interaction on React Native	Cross-platform framework limits advanced accessibility features, requiring custom plugins for full focus-change actions	Empirically evaluated	
EyeMath (Hernández-Sabaté et al., 2016)	Segments images into mathematical and plain text parts, creates Abstract Syntax Trees (ASTs) for math expressions	Recognizes text and mathematical symbols using OCR and MathPix API; converts LaTeX to Thai syntax	Supports navigation through image selection, history, and screen reader voice feedback	Compatible with TalkBack for Android, facilitating access to Thai mathematical content	Accuracy depends on spacing in images and OCR font limitations; complex nested scopes in math expressions may reduce clarity for users	Empirically evaluated	
Math Reader (dos Reis & Lorenzi, 2020)	Utilizes structural, lexical, and syntactic analysis to interpret handwritten math expressions	Recognizes handwritten symbols and notation using CNN and additional algorithms for graphic and structural analysis	Web application interface for math input; facilitates accurate expression input and interpretatio	Primarily a visual tool for web use, which may require adaptation for full compatibility with screen readers	Accuracy depends on handwriting clarity and training data; limited accessibility without screen reader adaptations	Not empirically evaluated	
CAVI (Mejía et al., 2021b)	Designed for visually impaired users, offering algebraic and calculus operations via Maxima engine	No	Sequential navigation with keyboard shortcuts and spoken feedback	No, relies on auditory feedback rather than visual elements	No detailed navigation or equation structure feedback, focused on solution only.	Empirically evaluated	
MathCAT (Verschoor & Straetmans, 2010)	Provides accessible MathML support for visually impaired users, parsing mathematical structures for speech output	Identifies elements like operators, numbers, and variables in expressions	Supports structural navigation through expressions	Integrates with major screen readers for compatibility with auditory cues and braille devices	Limited to MathML-supported formats; requires updates for more complex or newer mathematical notation standards.		
EuroMath (Fitzpatrick, Nazemi & Terlikowski, 2020)	A web-based platform facilitating mathematical communication for teachers and visually impaired students, enabling creation and access to content in multiple formats	Identifies mathematical content, including text, equations, and graphics	Allows navigation through a freeform editor with various input modalities (braille, speech, large print)	Compatible with major screen readers and assistive technologies, ensuring accessibility for various users	Requires familiarity with the platform; potential barriers in technology access for some students	Empirically evaluated	
IrisMath (Zambrano et al., 2023)	Supports both linear and nonlinear equations with layered architecture for complex expressions.	Identifies components of equations (variables, operators, etc.) effectively.	Intuitive interface with visual and audio cues for navigation.	Yes	Requires internet access; still under development for user adoption.	Empirically evaluated	
CAS (Computer Algebra System) including Mathematica, Maple, and Maxima (Nieto & Ramos, 2021)	Provides rich symbolic and numeric computation, suitable for complex mathematical tasks.	Can identify and parse various mathematical elements, but may require adaptation for auditory feedback.	Offers extensive keyboard shortcuts but can be complex to navigate without visual aids.	Has screen reader compatibility, but advanced features may not be fully accessible.	Limited navigation. Inaccessible interface, steep learning curves, inaccessible graphics, and inaccessible output for blind students	Not empirically evaluated	
Desmos Math calculator (https://www.desmos.com/accessibility#overview)	Supports both visual and auditory representation of equations, allowing students to hear descriptions of graphs and equations, aiding in structural comprehension.	Identifies key elements within equations, such as variables, functions, and operators. Desmos’ accessibility mode highlights terms and points on graphs.	Provides keyboard navigation shortcuts, enabling students to navigate between graph elements, equations, and settings	Compatible with popular screen readers, including JAWS, NVDA, and VoiceOver. Offers auditory feedback for equations, graphs, and elements like coordinates.	Lacks full support for advanced graphing features and some complex notations, which may limit its utility in higher-level mathematics.	Empirically evaluated	
Proposed solution	Provides a detailed breakdown of mathematical expressions into components such as terms, factors, operators, and grouping symbols. This structure facilitates better comprehension	Identifies key elements within equations (e.g., variables, constants, operators) and their relationships. Enables users to understand the role of each component.	Offers intuitive navigation through expressions, allowing users to move between terms and operators easily.	Yes	May not support all advanced mathematical functions or complex notations, limiting usability in higher-level mathematics.	Empirically evaluated	

Table 1 comparing the tools demonstrates that although current technologies offer different degrees of assistance when it comes to navigation, structure overview, and comprehension of mathematical equations, there is a lack of an effective, easy-to-use system that takes into account the multiple needs of students with visual disabilities.

Altogether, math content is complex and notated so screen readers cannot accurately convey spatial and hierarchical relationships. Conversion tools for MathML and LaTeX have challenges in formatting, navigation, and platform support (Maćkowski et al., 2023b). Although MathCAT is an interactive tool, it suffers from compatibility problems and lacks the flexibility to solve complex problems. Despite providing tactile access, braille displays and e-books do not support complex mathematical expressions and require braille proficiency (Neuper, Stöger & Wenzel, 2022). Due to their complex interfaces and outputs, computer algebra systems such as Mathematica and Maple are largely inaccessible. In comparison to our proposed solution, which incorporates, interactive navigation through keyboard input with speech feedback, collaborative support, semantic parsing of mathematical expressions, and broad accessibility via Python-based implementation, the existing multi-sensory augmented reality (AR) system is limited by region-specific braille inconsistencies, reliance on specialized hardware, lack of collaborative functionality, absence of semantic structuring for mathematical content, and limited empirical validation (Mikułowski & Brzostek-Pawłowska, 2020). The proposed solution addresses this issue and designed an algorithm for presenting a structured overview of mathematical expression with a flexible navigation key for comprehension to students with visual disabilities.

Proposed solution

The student with visual disabilities encounters numerous problems in learning mathematical expressions and equations. Existing tools and methods provide only syntactic representations of equations, which makes it difficult to fully understand how variables, constants, and operators are related mathematically. As a result of a lack of a structured semantic framework, students who use assistive technologies face barriers. To achieve this, a solution is needed that not only translates mathematical content into accessible formats but also retains the semantic depth of the expression.

The essential purpose of this solution is to deliver mathematical expressions in a well-structured and accessible context. Students with visual disabilities will benefit from the system's detailed semantic breakdown of expressions. The framework can also work with current assistive technologies, facilitating easy integration into existing educational environments and the simplicity with which the user navigates and interacts with equations. As a result, this solution presents more granular, organized, and editable representations of mathematical expressions when compared to existing tools. The proposed solution is primarily designed for linear and quadratic expressions, and its capabilities do not yet extend to more advanced mathematical concepts such as calculus or differential equations.

Architecture of the proposed solution

There are five essential modules in the proposed solution, the first converting mathematical expressions into content MathML. The second module parses the MathML content, the third module extracts semantic information, and the fourth module creates interactive keys. In the fifth module, the output speech is generated. Students with visual disabilities can use the solution to gain a deeper understanding of mathematical expressions’ actual meaning and structures. This makes it easy for users to navigate within the expression. This process involves making the proposed solution compatible with assistive technology such as screen readers, audio feedback, and results. Figure 1 provides a schematic explanation of the proposed solution.

Figure 1 Architecture of the proposed solution.

Mathematical expressions to content MathML

These expressions were then transformed into the Content MathML format by the use of a mathematical software called SymPy. With this purpose in mind, some key variables and associated symbols have been declared to be used in SymPy. Thus, plain text was chosen as an input format to eliminate the necessity to work with AsciiMath or LaTeX markup languages. Content MathML was obtained using the MathML function of SymPy and by setting printer=’content’. The generated MathML was checked to reflect the correct mathematical equations and could either be printed saved or copied into other applications that support precise mathematical expressions.

<math xmlns="http://www.w3.org/1998/Math/MathML">

 <apply>

  <plus/>

  <apply>

   <divide/>

   <cn>1</cn>

   <ci>x</ci>

  </apply>

  <apply>

   <power/>

   <ci>x</ci>

   <cn>2</cn>

  </apply>

 </apply>

</math>

Parsing content MathML

Using the lxml library, preprocessing of MathML input confirmed its valid structure before extracting math content. The XML input was interpreted into an element tree, which sorted variables, constants, and operators according to the element tree. To successfully move through the tree, nested operations and structures including addition, multiplication, and exponentiation used a recursive function. For an exact representation of complex aspects, such as fractions and nested expressions, structured components were created, which can also be ready for further processing or for improving accessibility. This method required the testing of a variety of mathematical equations to ensure its validation.

XML tree construction

The parsing process was initiated utilizing SymPy, a Python library intended for symbolic mathematics. To convert MathML representations into SymPy expressions, the sympy.parsing.mathml module used the parse_mathml() function. This function understood and correctly transformed the supplied MathML structure into a symbolic format that allowed SymPy to perform additional mathematical operations and analysis. This conversion then changes the input into an XML tree form where the form elements are placed in a hierarchy based on a tag such as, <math>, <apply>, <plus>, and other operators of the expression as shown in Fig. 2. The following are the tree representations of expression.

Figure 2 Representation of XML tree.

(1) 1x+x2.

<math> is the root node, which contains a single apply> element.

The <apply> element has three children: <plus>, another <apply> (for the fraction), and yet another <apply> (for the exponent).

The first child <plus> has no further children, making it a leaf node.

The second <apply> (representing the division) has two children: <cn> (with the value 1) and <ci> (with the variable x).

The third <apply> (representing the power operation) has <ci> (the variable x) and <cn> (the exponent 2) as children.

Sematic components extraction

After the XML tree structure is generated, an algorithm is employed to search the tree to determine certain components of the mathematical expression. These components are terms, factors, constants, variables, and relations. This helps break down each element and analyze the mathematical relations to make an orderly arrangement possible.

This extraction process ensures that the parsed output not only preserves the equation in a readable form but also maintains the hierarchical and semantic structure of the MathML content. Finally, the output is returned with the semantically required aspects so that it can be used for further manipulation, understanding, or depicting the mathematical expressions in a visual/audio format for easier learning or computational solving.

Algorithm 1 Parsing content MathML, extracting semantic components and generating feedback.

Input: a mathematical expression in content MathML	
1. The necessary libraries were imported, including keyboard, py-asciimath, smpy, pyttsx3, gTTS, and lxml.etree.	
2. The input mathematical expression was converted to a Content MathML expression using SymPy.	
3. The function parse_mathml (mathml: str) was defined in such a way that it accepted a Content MathML string as an input.	
4. The input was then parsed as MathML using etree. fromstring() method to generate an XML tree.	
5. Data Structures were Initialize. Empty lists were created to store: terms (for mathematical expressions).

factors (for multiplicative components).

constants (for numerical values).

variables (for symbols or identifiers).

relations (for inequalities or comparisons).

representations (for plain text descriptions of the components).

	
6. Recursive Function was Defined. a. to perform the traversal of the XML tree a nested function named traverse(element) was defined.

For each element encountered: i. The tag name was determined after parsing the tag string.

ii. Classification of Elements was Performed: • If the tag was ci, the variable content was added to the variables list.

• If the tag was cn, the constant content was added to the constants list.

• If the tag was applied, the operator was identified from the first child element: If the operator was divided, the words numerator and denominator were obtained and a string was added to the terms list in the following format: “Numerator divided by Denominator.”

If the operator was plus or minus, terms were extracted from child elements and appended to the terms list, with representations like “Term 1 + Term 2.”

If the operator was timed, factors were extracted and appended to the factors list, along with a representation like “Factor 1 * Factor 2.”

If the operator was power, the base and exponent were extracted, and a formatted string was appended to the terms list, expressed as “Base raised to the power of Exponent.”

If the operator represented a relational expression (e.g., lt, gt), left and right expressions were extracted, and a formatted string was appended to the relations list, expressed as “Left Expression < Relation > Right Expression.”

If the operator was enclosed in brackets, the nested structure was handled by recursively calling traverse() for the content within the brackets.

	
7. Recursive Traversal was conducted: a. The traverse() function was invoked for each child element to traverse the XML tree recursively.

	
8. Return Values were provided: a. After the traversal was complete, the function returned the populated lists of: i. Terms: Containing all identified terms in the expression.

ii. Factors: Containing all multiplicative components.

iii. Constants: Containing all numerical constants.

iv. Variables: Containing all identified variables.

v. Relations: Containing all relational expressions.

vi. Representations: This incorporates plain text descriptions of the mathematical structures.

	
9. Navigation keys were designed a. Navigating through terms was enabled by pressing Ctrl + T.

b. By pressing Ctrl + S, sub-term navigation was enabled.

c. By pressing Ctrl + F terms were enabled.

c. By pressing the left or right arrow keys, the cursor moved accordingly.

d. Exiting the program was possible by pressing Esc

10. The information displayed and the speech output is utilized a. User instructions and messages were displayed appropriately.

b. To provide auditory feedback, text messages were converted to speech using the pyttsx3 library.

	
Output: list of extracted terms, factors, total constants, variables, relations, structure overview, representation, and output feedback.	
End of algorithm	

Using lxml.etree.fromstring(), MathML content is parsed into an XML tree displaying its structure. Recursive traversal extracts terms, factors, constants, variables, relations, and representations as organized lists.

Design algorithm

The following are the steps of the designed algorithm.

The algorithmic flow for parsing mathematical expressions in Content MathML is depicted in Fig. 3. The following is the step-by-step procedure of the whole process, such as mapping an expression, parsing through the XML structure, getting semantic parts of the expression like terms and variables and last but not least providing navigational and auditory assistance.

Figure 3 Algorithm flow for MathML parsing and navigation.

The algorithm’s complexity is in three cases: best, average, and worst. In the best case, the input expression is simple and requires minimal operations, resulting in a time complexity of O (1). The average case depends on the size and complexity of the input expression, with nested loops and regular expression operations contributing to a time complexity of O (n). The worst-case scenario is a large and complex input expression, resulting in a time complexity of O (n2) due to numerous nested loops and the use of multiple lists to store intermediate results, which can also lead to increased memory usage.

Design navigation keys

The function of interactivity with the help of keys in the navigation was designed for the user with visual disabilities. With the help of the ‘keyboard’ library, and “pyttsx3”, library was able to handle ‘keypress’ types that let users easily move through mathematical equations. These keys enabled users to traverse the expression level, term level, term relation level as well as level of factors, constants or variables. When commands were entered into the system, it read back the needed piece of information in a spoken format. Features such as keyboard control, labelling for screen readers, logical flow of navigation, and simple auditory descriptions were maintained according to WCAG 2.1. A more detailed list of key functional areas within the detailed navigation is included in Table 2.

Table 2 Navigation keys and commands for comprehension of mathematical expressions.

Navigation key	Command	Description	
Read entire expression	Ctrl + E	Reads the entire expression loudly	
Read individual terms	Ctrl + T	Reads each term individually	
Read terms relation	Ctrl + R	Reads the relationship between terms	
Navigation	Ctrl + N	Provides options for navigating through the expression	
Navigate term by term	Left and Right arrow key	Allows the user to move through the expression term by term	
Read factors	Ctrl + F	Reads the factors of each term	
Read constants	Ctrl + C	Reads constants	
Read variables	Ctrl + V	Reads variables	
Exit	ESC	Exit the navigation mode	

The designed keyboard shortcuts provide a streamlined user experience, ensuring predictability through consistent Ctrl use and ease of navigation with left-right arrows. Adaptability offers shortcuts that fit diverse needs, while clear functions make task execution intuitive.

Output speech

The objective of this phase is to provide real-time speech feedback to enhance user understanding and engagement. Math expressions were converted into speech using a Text-to-Speech (TTS) engine, specifically the Python library gTTS (Google Text-to-Speech) navigation keys allowed users to explore individual terms, factors, and more with iterations through the whole expression, as well as adjust speech output settings.

Implementation of the proposed solution

The proposed was implemented using Python, in conjunction with the SymPy library which served as the core tool for symbolic mathematics. Additional XML parsing libraries were also used for traversing the Content MathML structure. With the implementation, users with visual impairments will be able to access the system using a screen reader and also text-to-speech facilities.

Use case of the proposed solution

Students with visual disabilities can learn mathematical expressions with the proposed system using this use case. Through auditory feedback, the system allows students to navigate through an interactive learning environment with flexible navigation based on mathematical expressions converted into Content MathML.

Actors: Students with visual disabilities are the primary actors in this use case, while educational instructors and support staff are secondary actors.

Preconditions: Students must possess a foundational understanding of assistive technologies and install the proposed solution on their devices.

Post conditions: Students gain an enhanced understanding of mathematical expressions after successfully interacting with the system.

(2) Themainflowofexpression2x+4=0isasfollows:

a. System initialization: The student power on the application. The auditory interface informs the client concerning the options of the functional capabilities, as well as how commands may be made.

b. Input of mathematical expression: The student types in the linear equation in the form 2x + 4 = 0 using a keyboard or a braille reader instead of a mouse. The system translates the given input into an intermediary form called content MathML.

c. Parsing the expression: The system will first analyze the equation to identify the semantic components of the equation such as 2x, the constant 4, and the relational operator =.

d. Interactive navigation: With an optimized keyboard key, the student navigates through the equation. The system provides vocal feedback for each term:

“Term 1: Two times x.”

“Term 2: Plus four.”

“Relational operator: Equals zero.”

e. Understanding structure: The student requests elaboration on 2x. The system responds with a detailed explanation: “Two times x means the coefficient 2 is multiplied by the variable x.”

f. Modifying expressions: The student independently chooses the coefficient 2 and substituted it by 3. The system updates the equation in real time and reads the updated expression: “Three times quantity x plus four amounts to zero.”

Materials and Methods

The proposed solution addresses the challenges faced by school students with visual disabilities in understanding mathematical expressions and equations. This solution includes an algorithm designed to align the mathematics curriculum with effective teaching methods. The algorithm parses MathML expressions, extracts semantic information, integrates interactive keyboard controls, and provides speech feedback to enhance user interaction with the application. The accessibility and effectiveness of this solution were validated among 15 instructors and 94 students with visual disabilities using Cronbach's alpha to measure internal consistency.

Participant’s demography

The study was conducted with 15 instructors eight male and seven female and 94 students 50 male and 44 female from government institutes for the blind at Peshawar, Abbottabad, and Swat Khyber Pakhtunkhwa Pakistan. The study involved students of both genders aged between 11 and 20 years and was focused on the grade 6 to 10 students. The participants were chosen deliberately to comprise a good sampling of the target group of students with visual disabilities. The sample size was deemed sufficient to assess the solution’s efficiency while addressing the concerns regarding the feasibility of the study and statistical significance. Demographic data are provided in Table 3.

Table 3 Participants’ information details.

Variable	Group	No of participants	Percentage	SD	
Gender	Male	58	53.211	0.0351	
Female	51	46.788	
Age	10–15	35	32.110	3.316	
16–30	35	32.110	
31–60	39	35.780	
Instructor or student	Instructor	15	13.762	7.817	
Student	94	86.239	
Type of visual disability	Blind	90	82.569	5.350	
Low vision	19	17.431	
Assistive technology usage experiences	6 Months	40	36.697	3.927	
1 year	25	22.936	
2 year	24	22.018	
4 year	20	18.349	

Research design

The solution was assessed by the Technology Acceptance Model (TAM) by Davis (1989) and this model measures the level of technology acceptance with perceived usefulness and perceived ease of use. The TAM framework is especially appropriate for this study since it is used to delineate its acceptance by the students and instructors who are part of the impaired vision population. Questionnaire data were collected concerning user experiences, and the TAM-based questionnaire used was reviewed and developed in consultation with a subject-matter expert in the field to ensure that the collected data are reliable. The reliability of the self-developed researcher-administered questionnaire was determined using Cronbach’s alpha, which established internal consistency.

Potential participants were explained the purpose of the study and any participant interested in participating in the study duly signed a consent form. After a 15-day experimental immersion in the proposed solution, they responded to the questionnaire, and the results showed a comparison to existing similar tools like NVDA and Access8Math.

Evaluation criteria and process

The proposed solution’s effectiveness was assessed based on the following criteria:

Accuracy: Tested participants’ knowledge on how to label terms, factors, constants and variables in mathematical expression.

Usability: Based on a TAM-based survey, found out the perceived ease of use (PEOU) and perceived usefulness (PU).

Comprehension: Evaluated according to respondent’s explanations and suggestions given after interaction with the system.

Navigation Efficiency: Controlled the time that it took and the number of mistakes made when trying to navigate through equations using an icon-like keyboard.

Random participants were supposed to use the specially designed system, so they went through the procedure to become trained with the system before the evaluation. They then applied the proposed solution on mathematics problems of different degree of difficulty and examined the results that came from it and other facilities like NVDA and Access8Math. Collected data measures included questionnaire data in the form of performance logs and error rates paired with interviews. Objective data related to various problem-solving levels were statistically examined using the ANOVA test, while the qualitative data were analyzed using the thematic analysis in order to establish usability problems and to provide recommendations.

Data collection and analysis

Measurement and data collection included accuracy measures, the time taken to complete the tasks, the rate of mistakes, surveys, and interviews with the participants. Descriptive statistics were used in analyzing quantitative data to determine if there was a significant difference in performance with increasing problem difficulty and open-ended responses were coded to generate themes by usability and participant recommendations.

Ethical considerations

Each participant was given information about the study and its aim, methods, and possible dangers involved in the study, and each of them signed an informed consent before they could participate in it. To ensure the security of the participants, their data was only shared between the researcher and the participant and all data was kept confidential. The research conformed to ethical guidelines for data collection that were provided by the University of Peshawar, Pakistan institutional review board reference No. 105 dated 15th May 2024.

Results and discussion

This section describes the empirical evolution of the proposed solution by reflecting based on the evaluation criteria. In the subsequent section, the findings of the study in light of qualitative and quantitative comparison are illustrated to note the applicability, efficiency, and consequences of the proposed solution in contrast with the NVDA and Access8Math tools. The proposed solution’s performance assessment involved 109 participants with visual impairments, including 15 instructors and 94 students. To ensure that the participants felt at ease using the system before the evaluation, they were taken through some training sessions. The data that were collected in the quantitative form included questionnaires, logs of performances, and error rates while qualitative data included interviews. Quantitative analyses employed included analysis of variance (ANOVA) to compare performance means across the different levels of problem-solving. This use of both quantitative and qualitative analyses gives a full picture of the solution’s quality, functionality, and impact, which will be presented in the subsequent parts of the study.

Quantitative analysis

Accuracy

The effectiveness of the proposed solution was evaluated based on its capacity to navigate through terms, factors, constants as well as variables of the mathematical expressions and formulas. From the results obtained it became clear that the proposed solution had an accuracy of 85% which compared well with 70% of NVDA and 65% of Access8Math as shown below in Table 4 and Fig. 4A. This can be evidenced as significant progress in the proper interpretation of mathematical context which will help in the advancement of improving learning among the students with visual disabilities.

Table 4 Comparison of the proposed solution with NVDA and Access8Math.

Tool	Accuracy (%)	Time (min)	Error rate (%)	Usability score (1–5)	
Proposed solution	85	10	5	4.5	
NVDA	70	15	15	3.5	
Access8Math	65	20	20	3.0	

Figure 4 (A) Comparison with NVDA and Access8Math; (B) frequency distribution of instructors’ feedback; (C) frequency distribution of student responses.

The results of the assessment of the accuracy show that the suggested approach offers higher recognition of mathematical characters as compared to the state-of-the-art tools enhancing the learning process for students with visual disabilities.

Usability

Usability was assessed with a Technology Acceptance Model (TAM)-based questionnaire that estimated perceptions of Perceived Ease of Use (PEOU) and Perceived usefulness (PU) of the proposed solution from the instructors’ and students’ perspectives. A vast majority of instructors (86. 7%) and students (82. 9%) said that the solution was easy to use and helpful, finding most of the survey items stating positive opinions on the solution’s usefulness quite acceptable (Tables 5, 6 and Figs. 4A–4C).

Table 5 Frequency distribution of answers to selected questionnaire items from instructors (n = 15).

Questions	Strongly agree	Agree	Neutral	Disagree	Strongly disagree	Total	
I discovered that this solution aided my students in comprehending math handouts, exercises, and exams.	14	1	0	0	0	15	
I believe that this solution could assist my students in focusing more on their math studies.	12	2	1	0	0	15	
I have found that this solution is particularly useful when solving math problems.	10	2	2	1	0	15	
I think that this solution has the potential to improve my students’ capacity for independent study.	10	4	1	0	0	15	
In my experience, this solution is user-friendly.	13	2	0	0	0	15	
I found that this solution navigates, and identifies all elements of mathematical expressions comparatively screen reader and access8math.	10	2	3	0	0	15	
I wholeheartedly endorse this solution and would recommend it to others.	12	2	1	0	0	15	
Total	81	15	8	1	0	105	

Table 6 Frequency distribution of students responses received (n = 94).

Questions	Strongly agree	Agree	Neutral	Disagree	Strongly disagree	Total	
I found that the solution assisted me in learning mathematics.	44	50	0	0	0	94	
Compared with access8math and screen reader, the proposed solution comprehends math resources and navigates and identifies structural elements of mathematical expressions better.	40	48	6	0	0	94	
I found that using this solution math has shortened the time required to learn math.	35	45	14	0	0	94	
I would consider this solution to be beneficial.	40	50	4	0	0	94	
I found that this solution is easy to use.	25	49	20	0	0	94	
I found that this solution enabled me to concentrate better while studying.	39	46	8	1	0	94	
This solution would be useful to me in studying math, and I would recommend it to others.	30	40	24	0	0	94	
Total	253	328	76	1	0	658	

Calculating Cronbach’s alpha for instructors’ questionnaires.

(3) α=(kk−1)(1−sumofvariancesofitemscoresvarianceoftotalscores)

where k is the number of questions.

In this case, k = 7, the variance of the total scores is 15.924, and the variances of the item scores are 0.067, 0.353, 0.972, 0.4, 0.124, 0.696, and 0.353.

Plugging these values into the formula, we get:

(4) α=(77−1)(1−(0.067+0.353+0.972+0.4+0.124+0.696+0.353)15.924)

(5) α=(76)(1−2.96215.924)

(6) α=(1.167)(15.924−2.96215.924)

(7) α=0.949

Therefore, Cronbach’s alpha for this questionnaire is 0.949, which indicates a high degree of internal consistency among the items. The student’s responses were also calculated accordingly for the following Table 6.

Cronbach’s alpha for a student questionnaire was calculated accordingly and found a value of 0.81.

Therefore, Cronbach’s alpha for this questionnaire is 0.91, which indicates a high degree of internal consistency among the items.

To examine the differences in the responses, ANOVA analysis was conducted to determine the significance of the differences in the scores obtained. The results concerning the instructors revealed that there were differences in mean responses concerning the different usability aspects (F = 11. 22, p < 0. 001), suggesting variability in the instructors’ perception of the usability of the solution. Likewise, in the analysis of student data, the ANOVA results also gave significant differences which means that there are indeed different opinions regarding the effectiveness of the said solution by having F = 4.005 and p = 0.006.

Comprehension

Focusing on the rationale of the proposed solution one of the major research questions of the study was constructed to stress the enhancement of students’ understanding of mathematical content based on the presented solution. Almost all of the 94 students who participated in the survey affirmed or strongly supported the statement that the solution assisted them in understanding the mathematical expressions and dealing with other mathematical content better in comparison with other tools. Also, improvements were noted in the overall skills to focus and understand mathematical materials based on the majority of positive responses mentioned in Table 6. These findings underscore the potential of the proposed solution to improve comprehension by providing a more effective, interactive method of learning complex mathematical concepts for students with visual disabilities.

Navigation efficiency

To assess navigation efficiency, the time that was taken in accomplishing the tasks and the mistakes made during testing the proposed solution as compared to NVDA and Access8Math were considered. The proposed solution offered better results with an average task completion time of ten minutes and an error rate of 5% as compared to 15 min and 15% for NVDA as well as 20 min and 20% for Access8Math as shown in Table 4. This is also evident from these results that the proposed solution has a better navigation system which enhances quicker and more precise ways of dealing with math content.

Qualitative analysis

Qualitative feedback was collected from instructors and students to gain insights into their experiences with the proposed solution. This data highlights the tool’s unique contributions to navigation, comprehension, and contextual analysis of mathematical expressions, in comparison to existing tools like NVDA Screen Reader and Access8Math. The qualitative data were categorized into key themes, including Ease of Navigation, Comprehension, Contextual Analysis of Mathematical Expressions, Comparison with NVDA and Access8Math, and Areas for Improvement.

• Ease of use: Several participants mentioned the proposed solution’s user-friendly interface and accessibility features. For instance, one instructor shared, “The proposed solution is simple to navigate, even for students with limited experience in technology. It was easy to integrate into our teaching routines”. A student commented similarly “Similarly, a student remarked, “I found it straightforward to access the lessons and solve problems without needing constant guidance”. The feedback from this suggests that the adoption was indeed helped by the intuitive design. But a few students suggested that an introductory tutorial or user manual for first time users would be appropriate.

• Improved focus and comprehension: The proposed solution was used both by instructors and students found a big improvement in both engagement and in understanding content. An instructor stated, “This solution provides a complete learning experience by integrating navigation and analysis of math expressions, something the other tools don’t offer in the context of mathematical learning”. Another comment from a student highlighted how the solution helped bridge gaps in their learning: “The proposed goes beyond just reading; it helps me navigate, understand, and analyze mathematical expressions, which makes learning math much easier compared to NVDA and Access8Math”. Other students also highlighted “The proposed solution helps me understand what the equations mean, not just read them”. These responses align with the quantitative findings of improved performance metrics.

• Areas for improvement: The overall feedback was positive, but some challenges were identified. Students expressed initial difficulty adapting to the tool’s format, with one student stating, “I needed extra time to get used to this new system.” Instructors suggested additional training sessions or technical support might help ease the transition. For example, one instructor proposed, “A brief training module for teachers and students would make the implementation smoother”.

The qualitative feedback shows that the proposed solution is capable of enhancing the performance of navigation, comprehension and contextual analysis over the already existing tools as stated in the literature review and also supports the mathematical learning of the visually impaired students. Participants noted the directions for improvement, yet there are plenty of innovative concepts regarding the use of the tool in a bigger picture. This study is an important contribution to accessible education, introducing a new tool to bridge the gap in learning mathematics for visually impaired students while acting as a consolidation of usability and comprehension concerns. Unlike other assistive technologies, this proposed solution is unique in targeting mathematical comprehension. At the very beginning, adaptation faces challenge and this is where robust training modules need to be for scalability.

Statistical evaluation

The instructor and student question responses were analyzed by Analysis of Variance (ANOVA) which yielded significant differences in mean responses. With regard to the instructor data, the p value was as follows 0.00000022; this gives a picture where the means of at least two groups are significantly different with an F value of 11.22 and degrees of freedom of 6 for between and 98 for within group variations. Likewise for the student data, test of between-subjects effect indicated the ANOVA test yielded an F value of 2.94 and a p-value of 0.00638, much less than 0.05 thereby suggesting significant between-group differences in the means. These results have enabled me reject the null hypothesis in both the circumstances a circumstance. According to the results obtained, 91.43% of instructors and 83% students perceived the proposed solution as easy to use, thus concluding that PEOU was high. But a few of the students complained that they encounter some difficulties in using the solution, and some said that it does not greatly improve the amount of time it takes for them to learn mathematics.

Discussion

As per the findings of this research, the proposed solution is quite beneficial in the enhancement of mathematical learning to the students with visual disabilities. The proposed solution has achieved a higher success rate of 85% when compared to the present tools NVDA-70% and Access8Math-65%. By improving the visibility and influencing the way that traditions refer to mathematical expressions, it improves learning.

Usability feedback was positive, with 91.43% of instructors and 83% of students finding the proposed solution beneficial and easy to use, supported by high internal consistency (Cronbach’s alpha: 0.81 and 0.91). The overall mean estimate of effect size was 0.94 for the instructor group and 0.91 for the student group. The system also enhanced task completion and reducible error rates, providing a faster and less erroneous way to engage with material.

Finally, combining qualitative and quantitative data gives a view of the proposed solution effectiveness. Measurable improvements in learning outcomes were demonstrated with quantitative data, and qualitative insights revealed the experiences for users, such as improved navigation, comprehension and engagement. Taken together, these findings show that the tool fills in gaps with traditional assistive technologies and provide useful guidelines for future improvements. This study focused on algebra within a specific context; future research should explore other mathematical domains and settings. A reliable, efficient, and portable solution for teaching mathematics to visually impaired students, the proposed solution could be expanded with future advancements and made compatible with other adaptive technologies.

Conclusions

This study addresses to fill the significant gap of students with visual disabilities in mathematics learning especially in the comprehension and solving of mathematical equations and expressions. The proposed algorithmic approach based on converting the input math expressions to MathML, parsing them to semantic components and presenting a structural viewpoint improves learning through an interactive navigation and speech output. Empirical evaluations in real educational settings demonstrated that the developed solution improves learning outcomes, with an accuracy rate of 85%, outperforming existing technologies like NVDA (70%) and Access8Math (65%). Both instructors (86.7%) and students (82.9%) expressed high satisfaction levels, validated by strong internal consistency (Cronbach’s alpha of 0.895 for instructors and 0.81 for students).

Qualitative and quantitative findings are integrated to show the effectiveness of the proposed solution in enhancing mathematical learning for visually impaired students. Quantitative results validate improved learning outcomes and qualitative insights offer depth by identifying strengths in navigation, comprehension and engagement. The proposed solution fills gaps within existing technologies while promising an accessible approach to user centered education. This research contributes to inclusive education by improving the accessibility and effectiveness of math-related learning for students with visual disabilities. The proposed solution is only applicable to linear and quadratic equations, which limits its applicability to a broader range of mathematical topics. Additionally, the integration with other superior assisting devices like haptic feedback has not been considered. In future works, this framework will be extended to an additional hierarchy of concepts in mathematics, including functions, more advanced formulas and relations, as well as incorporate the relevant novel technologies into the design for students with visual disabilities. This will in turn enhance the production of a more effective all-inclusive educational tool that fulfills their needs.

Supplemental Information

Supplemental Information 1 Students questionnaire feedback.

Supplemental Information 2 Research Questionnaire.

Supplemental Information 3 User guide for code.

Supplemental Information 4 Questionnaire responses of both students and instructors.

Supplemental Information 5 Qualitative feedback of students and Instructors.

Supplemental Information 6 Code.

Additional Information and Declarations

Competing Interests

Author Contributions

Ethics

Data Availability

The authors declare that they have no competing interests.

Amjad Ali conceived and designed the experiments, performed the experiments, analyzed the data, performed the computation work, prepared figures and/or tables, and approved the final draft.

Shah Khusro performed the experiments, authored or reviewed drafts of the article, and approved the final draft.

Tahani Jaser Alahmadi conceived and designed the experiments, performed the computation work, authored or reviewed drafts of the article, and approved the final draft.

The following information was supplied relating to ethical approvals (i.e., approving body and any reference numbers):

The university of Peshawar, Pakistan granted Ethical approval to carry out the study within its facilities (Institutional Review Board reference No. 105).

The following information was supplied regarding data availability:

The code is available in the Supplemental File.

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
