# Peer review of "Accessible interactive learning of mathematical expressions for school students with visual disabilities"

_PeerJ Computer Science, doi:10.7717/peerj-cs.2599_

## Round 0.1 · original submission · Major Revisions

Dear authors,

The reviews of your manuscript are included at the end of this letter. We ask that you make the necessary changes and additions to your manuscript based on these concerns and criticisms. Furthermore, equations should be used with equation number. Explanation of the equations should be checked. Definitions and boundaries of all variables should be provided. Necessary references should also be given. The equations are part of the related sentences and special attention is needed for correct sentence formation.

Best wishes

Reviewer 1 ·

Basic reporting

The scope of the paper is very novel and very important for teaching and learning of people with visual impairment.

The process and the evaluation and the number of participants and its type seems adequate

However I was lost on the findings sections, not only we can not see the figures and tables as we are reading the session (only in the end), the tables and figures at the end seem quite vague and without any detailed description on the findings secions.

The evaluation of the study should add some qualitative date analysis to understand the answeres and the figures. The findings show add the errors on each of the bars example figure 4.

Overall I think that the authors have more data and comparisions to present based on the evaluation process , than those they described. I suggest the authors to have a more deep analysis of the questionnaires and rational behind the selection.

Experimental design

Adequate.

Validity of the findings

Not enought, need to have a deeper analysis and add quotes and qualitative date to be more support the findings.

Reviewer 2 ·

Basic reporting

The article raises an interesting topic related to providing opportunities for blind students to work with mathematical formulas. The language is clear and readable. The paper contains a rich literature review; however, the authors did not mention several more complete systems supporting mathematics teaching, such as Euromath, Desmos, or Lambda. After reading the article, a few more detailed comments come to mind.
1. In section 2.1, the authors mentioned various screen reader tools that support mathematical expressions and added references to works describing these tools. However, the text must include their names and briefly present their functionality. I propose supplementing this section with information about the names of these tools, their basic capabilities, and their imperfections regarding the proposed solution.
2. The article's structure is correct, although some chapters seem too short, and, in my opinion, a 3-level structure is not necessary in this case. For example, I suggest not to divide section 4.3 into level 3 subsections in which each subsection contains one sentence. In addition to this, describe the evaluation criteria in one subsection.
3. A similar comment about section 4.4 comes to mind. I suggest not dividing it into subsections and describing the evaluation design in one subsection.
4. Table 1, in its current form, needs to contain sufficiently accurate information but gives only a general comparison of the tools with the proposed solution. What tools exactly do the authors mean? Suppose it is to be a comparison with existing tools. In that case, it should include specific tools and their features compared with the features of the proposed solution, e.g. Mathplayer does not allow viewing the structure of the equation, but the proposed solution allows for this. For a clearer understanding of the topic, it would be better to see some specific, more popular tools compared in this table with the proposed solution.

Experimental design

The article's content is original, and the topic discussed is important. Although similar research exists, the proposed solution can be considered a continuation, development, or a different approach to the issue. However, the method description would require supplementation and minor tidying up.
1. In section 2.6, some information from previous sections is repeated. I suggest checking the text for these unnecessary redundancies.
2. In section 3.1, the authors described using the SymPy library to convert a math equation to a MathML format. However, the question arises here as to what the input format of this equation was, whether it was, for example, Asciimath, Latex, braille code, or some other format.
3. Section 3.2 needs to be clarified. It needs to describe what is involved in parsing the input expression and extracting variables, numbers, operators, etc. A description explaining how it works would be helpful. The authors only mention using the Sympy library, which seems too poor.
4. In section 3.6, the authors described how the program communicates with the user using synthetic speech. However, there needs to be more description stating how variables, constants, and operators extracted from a mathematical expression are transformed into text that will then be read to the user. How are such formula elements considered, such as the beginning and end of the numerator in a large fraction or where an expression in brackets begins and ends? It would be helpful to have an example of how to read a more complex expression containing parentheses, expressions in indexes, generalized sums and products, generalized limits, or multi-story fractions.
5. In sections 4.1 and below, the authors mention that 94 students and 15 instructors participated in the study. However, in section 5, we find 105 blind respondents. I suggest clarifying this information. Additionally, it would be interesting to find out whether the instructors were sighted or if blind teachers were among them.
6. The strength of the article is that the evaluation of the solution by users and statistical research with data from 94 students and 15 instructors is described precisely and comprehensively. The authors attached much data they used during their research to the article. Surveys were conducted among users, tables and drawings explaining the described experiment and the code of the developed software. The only inconvenience with this part of the submission is that to check the operation of the code; a reviewer needs to install additional libraries and then compile the solution. At this stage, it would be helpful to have instructions on how to run it or, even better, a compiled version of a working solution. For this reason, I could not check an exciting solution and verify how it works with the screen reader and keyboard.

Validity of the findings

1. Although the topic discussed in the paper is undoubtedly important, the solution described is simple and cannot be considered entirely innovative. Similar solutions using Asciimath to MathML conversion and generating semantic text content describing formulas for a blind user have been implemented in projects such as Euromath, Desmos, and Lambda. However, the described solution is a good continuation and complement to these projects.
2. The Summary Section lacks information about what the authors intend to do with the proposed solution in the future or whether they plan to develop, improve, and implement it.

Additional comments

The solution described in the article can undoubtedly help a blind student learn mathematics by exploring mathematical formulas, as the authors proved by conducting research among students and instructors. However, it is known that most of the work with mathematics during lessons involves solving tasks independently. Therefore, extending the solution described in this article is worth extending so that students can read, create, and edit their formulas.

---

## Round 0.2 · Minor Revisions

Dear authors,

Thank you for the revision. One of the previous reviewers did not respond to the invitation for reviewing the revised paper. According to the other reviewer, your paper still needs to be revised and we encourage you to address the concerns and criticisms of the reviewer and resubmit your paper once you have updated it accordingly.

Best wishes,

Reviewer 2 ·

Basic reporting

The solution described in the article can undoubtedly help a blind student learn mathematics by exploring mathematical formulas, as the authors proved by conducting research among students and instructors.
The article's structure is meticulously crafted, ensuring a comprehensive and accurate presentation of the research.
The significant benefit of working is that the authors proposed references to reach a set of papers.
Despite its complete nature, the work has one limitation that I suggest supplementing and correcting. They are mainly concerned with existing tools available to support blind students' mathematics teaching.
Although Table 1 clearly compares the solution proposed by the authors with existing tools, the data in it seems not entirely true. Perhaps the authors did not find the latest works describing these tools.
For example, the Desmos Graph calculator has the ability to describe graphics using sound, and this is its main function
Similarly, the Lambda System has the ability to view the structure of the equation as well as identify the elements of the equation
Euromath system also has functions such as equation structure overview, Equations, or expression elements
identification, as well as expression Navigation
Therefore, I suggest adding to section 2. the description of the most advanced existing tools that are closest to the solution proposed by the authors. Example sources:

1. Mikulowski D.1 Brzostek-Pawlowska J.2 Multi-sensual Augmented Reality in Interactive Accessible Math Tutoring System for Flipped Classroom International Conference on Intelligent Tutoring systems, (2020),
2. https://www.lambdaproject.org/
3. https://veia.it/en/lambda product

Experimental design

The method, course of the algorithm, and its implementation are described with precision and clarity, making it easily understandable for the reader.

Validity of the findings

No comment.

Additional comments

No comments.

---

## Round 0.3 · Minor Revisions

Dear Authors,

Thank you for the work for the revised paper. Reviewers have now commented on your article and according to one reviewer, it is still not recommended that your article be published in its current format. We advise you to revise the paper in light of that reviewers' reviewers comments and concerns before resubmitting it.

Best wishes,

Reviewer 1 ·

Basic reporting

The new version is still missing a detailed evaluation of the different approaches. And some qualitative analysis of the comments of the students and educators.

Experimental design

nothing to add

Validity of the findings

it is still with lack of details. You should add more qualitative data on the different approaches.

Additional comments

I still miss the details on the evaluation, I suggest minor changes to give authors the possibility to improve their findings and discussion session, however due to the relevance of the topic to support students with visual impairment I will increase my score.

Reviewer 2 ·

Basic reporting

The authors propose a practical and effective solution that can assist blind students who speak English in learning mathematics. In addition to describing its method, implementation, and evaluation, the article also presents in section 2 a rich description of all existing tools that may be helpful in learning mathematics by blind people. The overview encompasses a wide range of essential tools, including screen readers and various tools related to MathML and conversion between different mathematical notation formats. The authors also presented more complete systems supporting mathematics teaching Hhalong and their advantages and disadvantages. This overview is illustrated in a table, which clearly shows the features of individual tools and compares them with the proposed solution. Due to this, in addition to presenting the author's solution, the article in this version can be considered a source of knowledge about the problems and solutions blind students encounter in teaching mathematics. This can be considered an additional advantage of the work.

Experimental design

Comments provided in the previous review and taken into account by the authors.

Validity of the findings

Comments provided in the previous review and taken into account by the authors.

Additional comments

Comments provided in the previous review and taken into account by the authors.

---

## Round 0.4 · accepted · Accept

Dear Authors,

Thank you for the revised paper. The previous reviewer declined to assess the latest revision, and I have therefore undertaken this task myself. It is my assessment that the paper has been sufficiently improved and is now ready for publication following the second revision.

Best regards,